# A novel vehicle-mounted sticky trap; an effective sampling tool for savannah tsetse flies *Glossina morsitans morsitans* Westwood and *Glossina morsitans centralis* Machado

Jackson Muyobela[1,2]*, Christian W. W. Pirk[1], Abdullahi A. Yusuf[1], Njelembo J. Mbewe[2,3], Catherine L. Sole[1]

**1** Department of Zoology and Entomology, University of Pretoria, Hatfield, Pretoria, South Africa, **2** Department of Veterinary Services, Tsetse and Trypanosomiasis Control Unit, Ministry of Fisheries and Livestock, Lusaka, Zambia, **3** Department of Disease Control, Faculty of Infectious and Tropical Diseases, London School of Hygiene and Tropical Medicine, London, United Kingdom

* u19395605@tuks.co.za

**Data Availability Statement:** All relevant data are within the manuscript and its Supporting Information files.

## Abstract

### Background

Black screen fly round (BFR) is a mobile sampling method for *Glossina morsitans*. This technique relies on the ability of operator(s) to capture flies landing on the screen with hand nets. In this study, we aimed to evaluate a vehicle-mounted sticky panel trap (VST) that is independent of the operator's ability to capture flies against BFR, for effective and rapid sampling of *G. m. morsitans* Westwood and *G. m. centralis* Machado. We also determined the influence of the VST colour (all-blue, all-black or 1:1 blue-black), orientation and presence of odour attractants on tsetse catch.

### Methodology/Principal findings

Using randomised block design experiments conducted in Zambia, we compared and modelled the number of tsetse flies caught in the treatment arms using negative binomial regression. There were no significant differences in the catch indices of the three colour designs and for in-line or transversely oriented panels for both subspecies (*P > 0.05*). When baited with butanone and 1-octen-3-ol, VST caught 1.38 (1.11–1.72; *P < 0.01*) times more *G. m. centralis* flies than the un-baited trap. Attractants did not significantly increase the VST catch index for *G. m. morsitans* (*P > 0.05*). Overall, the VST caught 2.42 (1.91–3.10; *P < 0.001*) and 2.60 (1.50–3.21; *P < 0.001*) times more *G. m. centralis* and *G. m. morsitans* respectively, than the BFR. The VST and BFR took 10 and 35 min respectively to cover a 1 km transect.

### Conclusion/Significance

The VST is several times more effective for sampling *G. m. morsitans* and *G. m. centralis* than the BFR and we recommend its use as an alternative sampling tool.

**Funding:** This work received financial support from the German Academic Exchange Service (DAAD) In-Region PhD Scholarship Programme 57511424 awarded to JM and, the South African National Research Foundation (NRF) Grant RA191211496819 to CP. The study was part of the postgraduate training programme of JM being undertaken at the University of Pretoria. The funders had no role in study design, data collection and analysis, decision to publish, or preparation of the manuscript.

**Competing interests:** The authors have declared that no competing interest exist.

## Author summary

The fly round is a mobile method used to sample *G. m. morsitans* and *G. m. centralis*, important vectors of human and animal African trypanosomiasis. However, its effectiveness is largely dependent on the skill and ability of the operator(s) to catch flies using a hand net. Here, we report the evaluation of an alternative mobile sampling tool, the vehicle-mounted sticky trap (VST) which is independent of operator skill and ability to catch flies. We show that VST is more effective in catching both female and male *G. m. morsitans* and *G. m. centralis* compared to the black-screen fly round (BFR). Furthermore, VST covered the same distance of BFR in a much shorter time. This study provides a basis for the use of VST in large scale sampling of *G. morsitans* to determine its geographical limit, a critical aspect in the planning of vector control strategies and interventions.

## Introduction

Tsetse flies (Diptera: Glossinidae) are the sole cyclical vectors of trypanosomes that cause human African trypanosomiasis (HAT or sleeping sickness) and animal African trypanosomiasis (AAT or nagana) [1]. Increased treatment and vector control in the last two decades has resulted in a substantial reduction in reported HAT cases, from over 30,000 in 1999 [2] to below 1000 in 2018 [3]. Despite this effort to eliminate HAT, the disease remains a considerable burden to rural communities [4]. AAT is a significant cause of poverty and malnutrition in sub-Sahara Africa and is estimated to cause three million cattle deaths annually [5]. This study focuses on two subspecies of the savannah tsetse fly *Glossina morsitans*, *G. m. morsitans* Westwood and *G. m. centralis* Machado. Both subspecies are efficient vectors of sleeping sickness and nagana [6], with HAT foci persisting within their geographic range [7].

Effective tsetse sampling tools are essential in the control of African trypanosomiases [8], as they provide critical information relating to distribution limits of tsetse populations, risk of trypanosomiasis to livestock and humans, and the outcome of vector control interventions [9]. Several trapping devices that exploit the host-seeking behaviour of tsetse, particularly, those related to visual and olfactory stimuli, have been developed. Trap development has been facilitated by the observation that *Glossina* can perceive shape, colour and movement [10]. Experimentation has shown that *Glossina* discriminates colour, with phthalogen blue (reflectance peak 450 nm) being the most attractive whereas black and UV-reflecting white stimulate landing [11–13]. The attractiveness of shapes increases in the order circle, square and rectangle [14], with horizontal rectangles being more attractive than vertical ones [15]. An array of traps with varying combinations of blue and black cloths have been developed for sampling *Glossina* [16]. Stationary traps developed for sampling *G. morsitans* include the NGU [17,18] and NZI [19] traps used in East Africa, and the epsilon trap recommended for use in southern Africa [20]. Pyramidal traps have recently been shown to be effective against *G. m. centralis* [15].

Host olfactory cues have been shown to increase trap catches of savannah tsetse species [21]. Components of host odour found to be attractive are carbon dioxide, acetone, butanone, 1-octen-3-ol, and several phenolic derivatives, such as 4-methlyphenol and 3-n-propylphenol [9]. Butanone and 1-octen-3-ol are generally used to bait traps during surveys and monitoring operations in the field due to their relatively low cost [22,23].

Stationary traps provide a standardised system of sampling, catch a higher proportion of females and can operate throughout the tsetse's activity period [16], but their use has limitations. Leak et al. [16] highlights the following as the major disadvantages of stationary traps: (i) they need to be well constructed and maintained to provide a standard sample; (ii) their

effectiveness is dependent on their deployment site; (iii) they give a misleading picture of daily activity rhythm [24]; and (iv), they are not sensitive enough to detect readily low-density populations of certain species such as *G. morsitans*. Furthermore, their high cost of deployment over large areas tends to limit their use in large scale surveys [9,16,25,26]. Consequently, there has been limited geographical sampling of natural tsetse populations [25], which hinders the understanding of several aspects of their population structure such as genetic sub-structuring [27].

Translocation has been demonstrated to significantly increase the attractiveness of an object to tsetse [28] and mobile baits are more effective in trapping *G. m. morsitans* than stationary baits [18,29]. Mobile baits used for sampling *G. morsitans* include the vehicle-mounted electric target [30] and black-screen fly round (BFR) [31,32]. The use of vehicle-mounted electric targets has mainly been limited to research studies as they are expensive and require high levels of maintenance [16]. Black-screen fly rounds, consisting of a catching party of two individuals with hand nets and a baited 1 × 1.5m black screen, produce samples that depend on the varying ability of people to catch tsetse flies with hand nets [16].

Therefore, the development of an effective, mobile sampling device that is not influenced by the operator's ability to capture *G. morsitans*, particularly at low population densities, is desirable. In this study, we evaluated a VST for effective and rapid sampling of *G. morsitans*. Specifically, the aim was to identify the optimal trap colour and orientation, and assess the need for olfactory attractants. Further, we evaluated the effectiveness of the VST against the BFR, for sampling *G. m. morsitans* and *G. m. centralis*.

## Materials and methods

### Ethical statement

Permission was granted by the Departments of National Parks and Veterinary Services Zambia to undertake entomological sampling in the game management areas. This study was conducted in conformity with the University of Pretoria ethical rules for animals.

### Study sites

The study was conducted in Zambia, between the longitudes 22 and 34˚E, and latitudes 8 and 18˚S. *Glossina morsitans* is the most widely distributed tsetse in Zambia covering an estimated 277,000 km$^2$ (or 38% of the total landmass). *Glossina m. morsitans* occupies the hotter eastern part while *G. m. centralis* occurs in the cooler western and Northern parts [33]. Studies were conducted at two sites, one for each subspecies. A brief description of each site is given below.

**Luano game management area.**   Mopane woodland is the dominant vegetation with small pockets of farmland, used mainly for growing maize and groundnuts. The livestock population is low with goats and chickens being the main species. Wildlife is abundant and represents the major blood meal source for the resident *G. m. morsitans* [34]. Experiments were conducted over 9 days in July 2020 along a 13 km motorable track north-east of Lubalashi village.

**Mumbwa South game management area.**   Miombo woodlands interspaced with large riverside dambos (grassy wetlands) is the dominant vegetation. No human settlements occur within a 50 km radius of the study site. Wildlife is diverse and abundant and represents the major blood meal source for *G. m. centralis* [35]. Experiments were conducted over 9 days in September 2020 along a 16 km motorable hunting track.

### Trapping devices and materials

Two fabrics were used in trap construction. These were blue polyester (PermaNet, Vestergaard Frandsen, Denmark) and 100% polyester black (225 g/m$^2$, Q15093 Sunflag, Nairobi).

They were cut and fixed on to one side of a 5 mm × 1 m × 1.5 m plywood board to produce a 1 × 1.5 m all-blue, all-black and 1:1 blue-black panel. A 5 cm strip of black duct tape was placed around the edge of the panel to secure the cloth onto the plywood board (Fig 1). A 0.5 × 1.5 m of the same 100% polyester black cloth was used to make the black screen used in BFRs.

To enumerate flies landing on trap panels, one-sided adhesive film (30 cm wide rolls, Rentokil FE45, Liverpool, UK) was fastened above the cloth, using black duct tape (Fig 1). Spectral reflectance of the cloth is not affected by the adhesive film except in the ultra-violet (UV) spectrum [15]. Virtually all UV wavelength below 380 nm is absorbed due to the addition of a UV absorber in the glue. Spectrophotometric measurements of light reflected from the adhesive film applied to fabrics similar to those used in this study indicated that all wavelengths in the UV range were absorbed by the fabrics [36]. New adhesive film was used at the start of each experiment.

Butanone and 1-octen-3-ol were used as attractants according to methods described by Torr *et al*. [37]. A 500 ml glass bottle with a 2 mm aperture in the stopper was used to dispense butanone at a rate of 150 mg/hr, while polyethylene sachets of 4 × 5 cm 500 gauge/0.125 mm dispensed 3 g 1-octen-3-ol at 0.5 mg/hr.

A 1 × 1.5 m rectangular steel frame with horizontal support legs (Fig 1) was used to mount the sticky trap panel on the back of a Nissan Hardbody, twin cab, 4×4 vehicle. To construct a two-sided sticky trap, two panels were fixed on either side of the steel frame. The same driver was used throughout the study. The rectangular trap panel was set up such that its longest side was horizontal to the ground in all experiments. Non-stick baking paper was used to cover the sticky surface of the trap when not in use (Fig 1).

## Experimental design

In all experiments, randomised block designs [38] were used to compared treatment effects. Different blocks constituted 1 km transects, set 300 m apart [39], with groups of adjacent days as experimental units [40]. Treatments were randomly allocated to days within blocks (S1 Appendix) such that each transect was traversed at the same time, in consecutive days. Thus, site and time of sampling variation was blocked.

The vehicle traversed transects hourly with closed windows at a maximum speed of 10 km/hr following recommendations for vehicle electric nets [41], from 7:00 to 17:00 hr for *G. m. centralis* and 9:00 to 17:00 hr for *G. m. morsitans*. This resulted in 11 and 8 experimental replicates respectively. The average time taken to complete a 1 km transect was 10 min. A 60 s waiting period was undertaken at the end of the transect to allow the trapping of trailing tsetse before milking of the trap. Trapped flies were removed from the sticky film with forceps, killed by squeezing the thorax, identified, sexed and enumerated.

After sampling, the vehicle was moved to the starting point of the next transect, with the sticky surface covered with non-stick baking paper. On average, a 25 min waiting period was undertaken before the commencement of a subsequent transect. This prevented the entry of flies from one transect into another, thus ensuring the independence of blocks.

In the colour experiment, one panel of the all-blue, all-black and blue-black was randomly placed at the left side of the steel frame in in-line orientation, to compare the mean catch of treatments. Tsetse catches on the blue and black sections of the blue-black panel were recorded separately to facilitate their comparison. Two panels of the most effective colour were placed on either side of the steel frame in the orientation experiment, comparing in-line and transverse orientations (Fig 1). Data for the left and right (in-line orientation), and front and back (transverse orientation) was recorded separately to compare one versus two-panel traps.

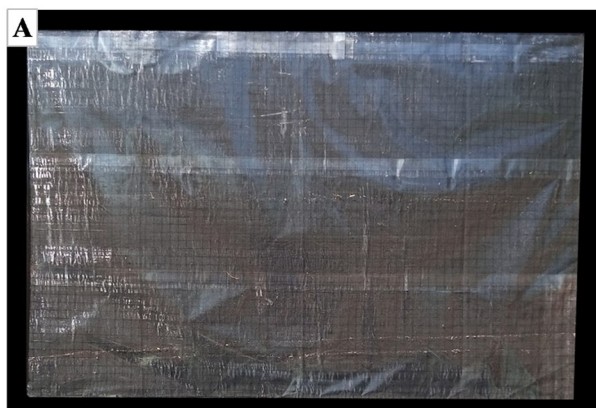
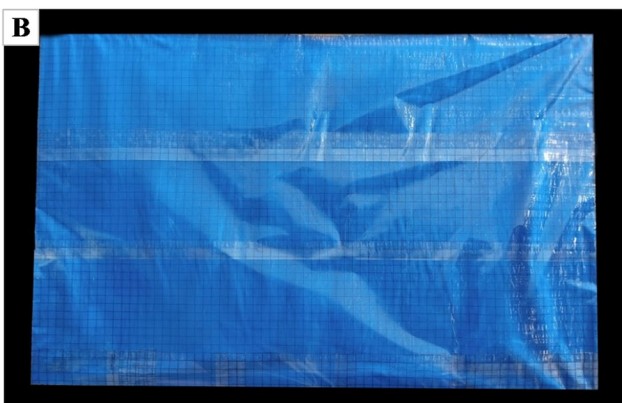
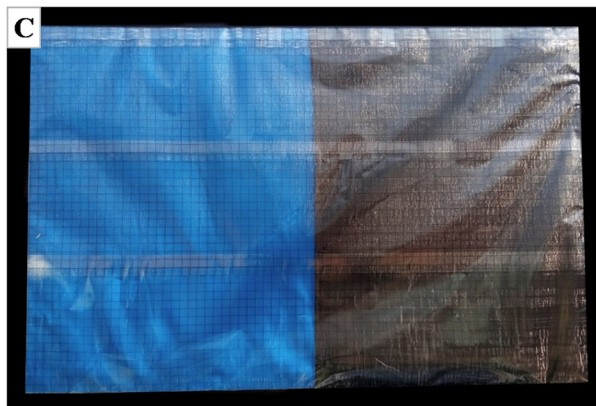
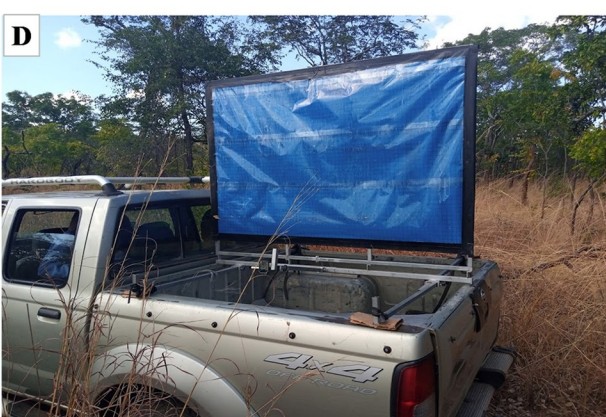
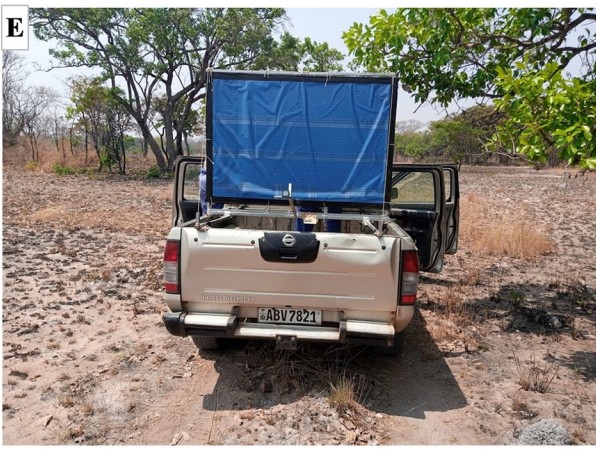
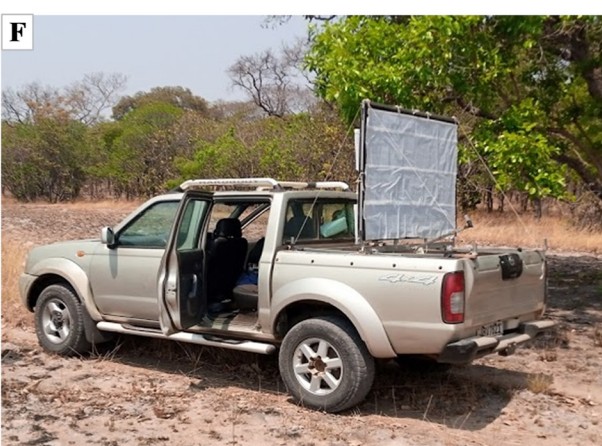

**Fig 1. Vehicle-mounted sticky panel traps.** (A) All-black sticky panel trap. (B) All-blue sticky panel trap. (C) 1:1 blue-black sticky panel trap. (D) All-blue sticky panel trap mounted in-line orientation (E) All-blue sticky panel trap mounted in transverse orientation (F) All-blue sticky panel in transverse orientation covered with non-stick baking paper.

Octenol sachet was fixed at the top while butanone was placed at the bottom of the leading edge of the trap to compare a baited and un-baited trap in olfactory experiments. When moving from one transect to another, odour dispensers were sealed in airtight containers and were reopened at the commencement of the next transverse.

The effectiveness of the optimised VST was finally compared with the BFR in a similar design. BFR was conducted as described by Robinson [31], where the catching party consisted of two men, each with a hand net, carrying a $0.5 \times 1.5$ m black cloth suspended from a 2 m pole and baited with octenol and butanone. The two men selected for the study had a combined 65-year experience in operating BFR and manning tsetse pickets. In each transect, 5 stops, 200 m apart, were identified. The catching party walked at normal speed to each stop where they caught tsetse flies that had landed on the screen or each other. After 3 min, the catching party moved to the next stop. On average, it took the catching party 35 min to traverse each transect. As the efficiency of BFRs reduces at high tsetse densities [16], this experiment was done in transects with low tsetse densities.

## Statistical analysis

To analyse changes of fly catches in the treatments, a negative binomial model with a log link was used in R [42]. The general formula of the model used was:

$$\log(\mu) = In(t) + B_0 + B_1x_1 + B_2x_2 + B_3x_3,$$

with μ representing the mean, $In(t)$ the dispersion parameter, $B_0$ the intercept, $B_1x_1$ the treatment, $B_2x_2$ the block and $B_3x_3$ the day effects of the model. Thus, mean tsetse catches were modelled on treatments for each experiment whilst taking into account the block and day effects of the study [38]. To provide a common index of effect, the mean tsetse catch of a treatment was expressed as a proportion of a reference treatment or control, the resultant value termed catch index. Model coefficients were used to estimate catch indices of treatments, which were compared using the likelihood ratio test. The significance of each comparison was determined when the number 1 was not included in the 95% confidence interval (CI) of the ratio test. Catch indices of 2 or 0.5 indicated that a treatment caught twice or half as many flies as the control. The "*effects*" package in R [43] was used to obtain de-transformed means of treatments. All reported estimates are accompanied by 95% CI and the alpha level was set at 0.05 for statistical significance. Computations were conducted using R [42].

## Results

A total of 11,342 tsetse were captured during the study (Table 1) with 29% being females. Only the two *G. morsitans* subspecies, *G. m. morsitans* and *G. m. centralis* were collected. For both subspecies, no transect with zero catch in all sampling hours was recorded (S2 Appendix).

### Colour comparison

As shown in Table 2 and 3, overall catch indices of the all-black, all-blue and blue-black panels for both *G. m. morsitans* and *G. m. centralis* did not differ significantly. For female *G. m. morsitans*, the catch index of the all-blue panel was significantly lower than the all-black panel, catching 39% fewer flies. Catch indices for males of all three panels were not significantly

**Table 1. Total tsetse catches for each experiment segregated by sex.**

| Subspecies | Experiment | Female | Male | Total |
|---|---|---|---|---|
| *G. m. morsitans* | Colour | 170 | 730 | 900 |
| | Orientation | 340 | 1,068 | 1,408 |
| | Olfaction | 707 | 1,514 | 2,221 |
| | BFR vs VST | 176 | 681 | 857 |
| *G. m. centralis* | Colour | 347 | 912 | 1,257 |
| | Orientation | 419 | 955 | 1,374 |
| | Olfaction | 987 | 1,984 | 2,973 |
| | BFR vs VST | 123 | 229 | 352 |
| **Total** | | **3,269** | **9,842** | **11,342** |

different. Male and female catch indices for *G. m. centralis* were not significantly different for all three panels (Table 3). In both subspecies, the catch index of the blue and black sections of the blue-black panel was significantly different, with the black section catching more male and female tsetse (Table 2 and Table 3). An all-blue panel was selected for use in all further experiments.

## Orientation comparison

Overall, male and female catch indices of sticky trap panels oriented in-line or transverse to the movement of the vehicle were not significantly different for both *G. m. morsitans* and *G. m. centralis* (Table 2 and 3).

## Comparison of one versus two panel design

The in-line orientation did not show a significant difference between the overall, male and female catch indices of the left and right trap panels for data collected throughout the day and for separate morning and afternoon analysis for *G. m. morsitans* (Table 2). Overall catch indices of the front trap panel in transverse orientation were significantly higher for data collected in the afternoon and throughout the day (Table 2), catching 170 and 72% more flies, respectively.

Left and right panel catch indices of the in-line oriented trap for data collected throughout the day were not significantly different for *G. m. centralis* (Table 3). A separate analysis of data collected in the morning and afternoon periods showed that the right panel caught 4.42 (95% CI: 1.39–14.10 and *P < 0.05*) times (or 342%) more flies than the left panel, in the morning. In the afternoon, the catch index of the right panel significantly reduced to 0.47 (95% CI: 0.27–0.81 and *P < 0.05*) times that of the left panel, catching 53% fewer flies. As shown in Table 3, the catch index of the front panel of the transverse-oriented trap was significantly lower than that of the back panel, with the front panel catching 41% fewer flies. Two-sided in-line oriented trap panels were selected for use in all subsequent experiments.

## Olfaction

No significant difference was observed between the overall catch indices of the un-baited and baited panel trap for *G. m. morsitans* (Table 2). For *G. m. centralis*, the overall catch index of the baited trap panel was significantly higher (Table 3), catching 38% more flies than the un-baited trap. The sticky trap panel used in the subsequent experiment was baited with octenol and butanone for *G. m. centralis* while that for *G. m. morsitans* was un-baited.

**Table 2. De-transformed means and catch indices as estimated by negative binomial regression for female and male *G. m. morsitans*.**

| Experiment | Treatment | Female | | | Male | | | Overall | | |
|---|---|---|---|---|---|---|---|---|---|---|
| | | Mean Catch (95% CI) | Catch Index (95% CI) | *P-value* | Mean Catch (95% CI) | Catch Index (95% CI) | *P-value* | Mean Catch (95% CI) | Catch Index (95% CI) | *P-value* |
| **Colour** | All Black (Control) | 7.71 (5.74–10.35) | 1 | | 22.90 (18.74–27.99) | 1 | | 30.20 (25.04–36.42) | 1 | |
| | All Blue | 4.68 (3.24–6.75) | 0.61 (0.40–0.92) | 0.018 | 28.84 (23.69–35.08) | 1.26 (0.98–1.62) | 0.073 | 33.67 (27.85–40.71) | 1.12 (0.87–1.42) | 0.373 |
| | Blue—Black | 5.64 (3.98–8.01) | 0.73 (0.48–1.10) | 0.133 | 27.90 (22.90–33.99) | 1.22 (0.94–1.57) | 0.122 | 34.25 (28.36–41.37) | 1.13 (0.89–1.43) | 0.299 |
| **Blue—Black** | Black (Control) | 2.31 (1.31–4.08) | 1 | | 19.65 (16.13–23.93) | 1 | | 27.62 (17.99–42.40) | 1 | |
| | Blue | 3.11 (1.87–5.19) | 1.34 (0.78–2.33) | 0.278 | 8.17 (6.15–10.84) | 0.42 (0.32–0.54) | 0.000 | 12.75 (8.08–20.12) | 0.46 (0.25–0.82) | 0.008 |
| **Orientation** | In-line (Control) | 7.75 (5.49–10.92) | 1 | | 27.24 (20.58–36.05) | 1 | 1 | 36.02 (21.69–46.87) | 1 | |
| | Transverse | 7.24 (5.10–10.27) | 0.93 (0.56–1.55) | 0.780 | 21.13 (15.79–28.28) | 0.78 (0.52–1.17) | 0.212 | 29.14 (22.24–38.18) | 0.81 (0.55–1.19) | 0.263 |
| **In-line** | Left (Control) | 4.87 (3.39–6.98) | 1 | | 21.60 (17.38–26.83) | 1 | | 26.79 (21.38–33.55) | 1 | |
| | Right | 6.08 (4.33–8.55) | 1.25 (0.92–1.71) | 0.158 | 23.44 (18.96–28.98) | 1.09 (0.85–1.39) | 0.518 | 31.99 (25.75–39.74) | 1.19 (0.92–1.55) | 0.179 |
| **Transverse** | Back (Control) | 7.73 (5.25–11.37) | 1 | | 14.23 (9.92–20.40) | 1 | | 23.22 (17.35–31.08) | 1 | |
| | Front | 7.64 (5.19–11.26) | 0.99 (0.62–1.58) | 0.962 | 32.58 (23.61–44.97) | 2.29 (1.50–3.51) | 0.000 | 39.98 (30.50–52.40) | 1.72 (1.22–2.43) | 0.001 |
| **Olfactory** | Un-baited (Control) | 36.82 (27.16–49.92) | 1 | | 71.92 (51.72–100.03) | 1 | | 110.11 (80.27–151.05) | 1 | |
| | Baited | 40.40 (29.80–54.78) | 1.10 (0.77–1.56) | 0.598 | 89.28 (64.30–123.96) | 1.24 (0.84–1.84) | 0.256 | 130.31 (95.04–178.65) | 1.18 (0.81–1.73) | 0.356 |
| **BFR vs VST** | BFR (Control) | 2.91 (1.56–5.42) | 1 | | 32.01 (21.93–46.72) | 1 | | 35.34 (24.20–51.62) | 1 | |
| | VST | 20.30 (14.73–27.99) | 6.98 (4.54–11.33) | 0.000 | 70.32 (49.39–100.12) | 2.20 (1.50–3.21) | 0.000 | 91.86 (64.55–130.71) | 2.60 (1.78–3.81) | 0.000 |

## Comparison of black screen fly round (BFR) and vehicle-mounted sticky trap panel (VST)

A significant difference between the overall catch indices of the BFR and VST was observed for both *G. m. morsitans* and *G. m. centralis* (Table 2 and 3). The VST caught 160 and 142% more flies than the BFR, respectively. VST female catch index for both subspecies was also significantly higher, catching 598 and 248% more female *G. m. morsitans* and *G. m, centralis*, respectively.

## Discussion

The results of the colour experiments indicated that there were no significant differences in the overall catch indices of all-blue, all-black and blue-black trap panels for both *G. m. morsitans* and *G. m. centralis*. This result suggests that the colours used in this study had an equal effect in eliciting close-range orientation towards sticky surfaces, an observation supported by the finding that blue and black surfaces are equally attractive to tsetse [12,44]. Differences in landing responses were observed on the blue-black panel with the black section having a higher catch index, a result also consistent with other findings [12,28,45]. Further research is

**Table 3. De-transformed means and catch indices as estimated by negative binomial regression for female and male *G. m. centralis*.**

| Experiment | Treatment | Female | | | Male | | | Overall | | |
|---|---|---|---|---|---|---|---|---|---|---|
| | | Mean Catch (95% CI) | Catch Index (95% CI) | P-value | Mean Catch (95% CI) | Catch Index (95% CI) | P-value | Mean Catch (95% CI) | Catch Index (95% CI) | P-value |
| **Colour** | All Black (Control) | 8.04 (6.29–10.28) | 1 | | 18.28 (14.78–22.61) | 1 | | 26.99 (22.09–32.99) | 1 | |
| | All Blue | 7.01 (5.15–9.55) | 0.87 (0.60–1.26) | 0.467 | 14.78 (11.32–19.31) | 0.81 (0.58–1.12) | 0.187 | 21.78 (17.05–27.82) | 0.81 (0.59–1.10) | 0.158 |
| | Blue—Black | 10.35 (8.05–13.32) | 1.29 (0.95–1.75) | 0.106 | 16.28 (12.70–20.86) | 0.89 (0.67–1.18) | 0.420 | 28.45 (22.81–35.48) | 1.10 (0.80–1.38) | 0.699 |
| **Blue—Black** | Black (Control) | 6.21 (4.67–8.26) | 1 | | 11.00 (8.72–13.88) | 1 | | 18.87 (15.32–23.24) | 1 | |
| | Blue | 3.81 (2.70–5.39) | 0.61 | 0.008 | 5.81 (4.39–7.69) | 0.42 (0.32–0.54) | 0.000 | 9.77 (7.59–12.56) | 0.52 (0.40–0.68) | 0.000 |
| **Orientation** | In-line (Control) | 11.37 (8.74–14.80) | 1 | | 26.86 (21.55–33.49) | 1 | | 39.70 (32.14–49.05) | 1 | |
| | Transverse | 10.05 (7.64–13.20) | 0.88 (0.65–1.20) | 0.435 | 20.64 (16.33–26.08) | 0.77 (0.58–1.00) | 0.051 | 30.53 (24.45–38.11) | 0.76 (0.56–1.00) | 0.049 |
| **In-line** | Left (Control) | 5.59 (3.27–9.56) | 1 | | 12.39 (6.99–21.95) | 1 | | 18.33 (10.67–31.48) | 1 | |
| | Right | 6.00 (3.53–10.19) | 1.07 (0.52–2.21) | 0.831 | 16.43 (9.37–28.83) | 1.33 (0.56–3.20) | 0.431 | 22.41 (13.12–38.28) | 1.22 (0.54–2.82) | 0.550 |
| **Transverse** | Back (Control) | 6.36 (4.39–9.23) | 1 | | 12.14 (8.22–17.93) | 1 | | 17.19 (11.48–25.74) | 1 | |
| | Front | 2.62 (1.63–4.22) | 0.41 (0.25–0.68) | 0.004 | 6.86 (4.46–10.56) | 0.57 (0.34–0.94) | 0.021 | 10.59 (6.59–15.62) | 0.59 (0.34–1.03) | 0.043 |
| **Olfactory** | Un-baited (Control) | 26.94 (22.08–32.86) | 1 | | 44.35 (36.23–54.29) | 1 | | 72.28 (60.56–86.27) | 1 | |
| | Baited | 37.03 (30.92–44.34) | 1.37 (1.08–1.74) | 0.009 | 61.70 (51.09–74.53) | 1.39 (1.10–1.77) | 0.006 | 99.93 (84.49–118.20) | 1.38 (1.11–1.72) | 0.004 |
| **BFR vs VST** | BFR (Control) | 2.87 (1.71–4.79) | 1 | | 6.14 (4.23–8.92) | 1 | | 9.73 (7.34–12.92) | 1 | |
| | VST | 9.99 (7.43–13.42 | 3.48 | 0.000 | 12.50 (9.22–16.94) | 2.04 (1.53–2.73) | 0.000 | 23.57 (19.25–28.84) | 2.42 (1.91–3.10) | 0.000 |

recommended to evaluate the effect of other colours, panel sizes and rates of movement on the mean tsetse catch of trap panels.

Since the three colour patterns used in this study had similar catch indices, we recommend the use of any of the three in the construction of VSTs for surveys whose objective is to identify the precise distribution limits and the main ecological niches of *G. m. morsitans* and *G. m. centralis*. For surveys whose objective is to establish the physiological age of the population using ovarian dissection, all-black and blue-black trap panels are recommended. In this study, the blue colour was chosen for subsequent experiments due to its potential for being used with imaging devices that may aid automated geo-referencing of catches, a key feature for monitoring low-density populations expected after vector control interventions.

Investigations on trap orientations indicated that there was no significant difference with the sticky panel traps oriented in-line or transverse to the horizontal axis of the vehicle for both *G. m. morsitans* and *G. m. centralis*. This finding may be attributed to the observed circulating behaviour of tsetse before they land on a visual bait [13,45,46]. The circulating behaviour of the following swarm may ensure that tsetse is equally available for capture no matter the orientation of the stick panel traps. In-line orientation was selected for use in subsequent experiments as it appeared to be more stable during sampling and non-sampling periods.

The catch indices of left and right panels were not significantly different for *G. m. morsitans*, even for separate analysis of morning and afternoon periods. However, left and right panel catch indices for *G. m. centralis* were significantly different for separate analysis of morning and afternoon periods with the panel facing the edge of the woodland, catching more tsetse than the panel facing the grassy dambo. The underlying cause of the observed contrasting response of the two subspecies was unclear since the circulating behaviour of tsetse before landing on a visual bait [13,45], is expected to make flies equally available for capture on both panel traps, regardless of habitat structure. Further investigation into this observation is warranted as it may highlight behavioural differences between *G. m. morsitans* and *G. m. centralis* that are yet to be established. The two-sided sticky trap panel was selected as the most effective design.

For *G. m. morsitans*, baiting with butanone and 1-octen-3-ol did not significantly increase the catch index of the VST. This result supports the work of Vale [28] who unequivocally demonstrated that tsetse were primarily attracted to mobile hosts largely in response to movement rather than odour. Tsetse close-range orientation towards a host has also been shown to be unaffected by synthetic odours [47] with carbon dioxide being the only host odour component that induces alighting response [48,49]. Therefore, our results suggest that visual factors are more important than synthetic olfactory cues in the effectiveness of the VST to trap *G. m. morsitans*.

Baiting with butanone and 1-octen-3-ol was however observed to increase the catch index of the VST for *G. m. centralis*. This result appears to suggest that close-range orientation to a host and/or alighting response in *G. m. centralis*, is influenced by synthetic odours contrary to observations made in *G. m. morsitans* [21]. Differences between the subspecies in response to a stimulus have been previously reported. Torr *et al.* [50] showed that very few *G. m. morsitans* were attracted to or landed on $0.25 \times 0.25$ m tiny targets relative to the 1 m$^2$ target. However, Byamungu *et al.* [15] observed that 0.5 m$^2$ targets were just as efficient as 1 m$^2$ target for *G. m. centralis*. It is unlikely that evolved differences in innate behaviour could account for the observed differences in response to odours, given the taxonomic classification of these two organisms as subspecies. More probable is the postulation by Vale *et al.* [51] that habitat geometry affects the effectiveness of odour attraction and the strength of the landing response among tsetse flies. Perhaps the differences between the geometry of the habitats of the two subspecies is sufficiently great to affect their responses to odours. Habitat effects are considered to cause the observed differences between savannah and riverine species in response to odour baits [21], but more needs to be done to assess what differences there might be between subspecies of *G. morsitans*. Presently, we recommend the baiting of the VST with butanone and 1-octen-3-ol for sampling *G. m. centralis* but not for *G. m. morsitans*.

For both *G. m. morsitans* and *G. m. centralis*, the VST was observed to have a higher catch index than the BFR, despite the former being operated at 10 km/hr and the latter at 5 km/hr. This result could be attributed to a combination of several factors. Firstly, the VST presented a larger moving target than that of the BFR. It has been previously demonstrated that larger moving objects are more attractive to tsetse than smaller ones [10,13,15]. Secondly, a proportion of tsetse attracted to the BFR was repelled by human odour which has been previously shown to have a repellent effect on tsetse [9,28]. As human operators were enclosed within the vehicle with closed windows, no such repellent effect could be present for the VST. Lack of repellence could also explain the higher female catch index of the VST as compared to the BFR since repellence to human odour is higher in female flies [10,28]. Thirdly, BFR catches are influenced by the hand netting ability of operators [16], whilst the catch size of the VST is consistent within the effective range of the sticky film. Therefore, our results indicate that the VST is better suited for sampling both subspecies of *G. morsitans* than the BFR. However, we

recommend further studies to establish the cost-effectiveness of the use of the VST in comparison with other traps.

In this study, the VST was operated at 10 km/hr following recommendations for vehicle electric nets [41]. However, VST may be more effective in sampling tsetse at other speeds. Therefore, we recommend further studies to determine the most optimal operation speed for the VST. Nevertheless, given that flight speeds of up to 14.4 km/hr have been observed in the laboratory [52] and mean speeds of 22 km/hr recorded in the field for *G. m. morsitans* [53], we suggest that the maximum operational speed of the VST should not exceed 20 km/hr. This speed should be efficient for sampling moderate to high-density populations of *G. morsitans* with the lower speed of 10 km/hr being better suited to low-density populations. Intermittent stops at 1 km intervals ensured that most flies following the VST were captured. This practice is recommended to facilitate georeferencing of the catch.

A significant limitation of the VST is that sampling is largely restricted to motorable tracks. Thus, VST is likely to be used as a complementary sampling device to existing tools. The results showed that the VST had a higher sensitivity to male tsetse. This apparent bias may make it an ideal tool for monitoring flight ability, survival and competitiveness of sterile males released during the implementation of the sterile insect technique. Furthermore, since males copulate with multiple females, a vehicle-mounted panel target of a similar design could prove an effective mobile delivery system of entomopathogenic fungi for biological control of *G. morsitans*. Investigations on the suitability of using the VST as a personal protection device and control tool during game drives are further recommended.

We conclude that the VST constructed using an all-blue, all-black and 1:1 blue-black panel, oriented either in-line or transversely and baited with butanone and 1-octen-3-ol (*G. m. centralis*), is a more rapid and effective sampling tool for *G. morsitans* than BFR. This makes it an invaluable tool in planning and evaluating area-wide suppression/eradication campaigns. We recommend its use in sampling large geographic areas to facilitate *G. morsitans* population studies currently limited by the inadequate sampling of natural populations.

## Supporting information

**S1 Appendix. Experimental Design.**
(DOCX)

**S2 Appendix. Raw data.**
(DOCX)

## Acknowledgments

We thank the Chief Tsetse Control Biologist, Mr Kalinga Chilongo and staff of the Tsetse and Trypanosomiasis Control Unit, Ministry of Fisheries and Livestock Zambia, for their administrative and technical support. Special thanks go to Mr Getson Sikombe, Mr Emmanuel Mweetwa and Mr Milner Mukumbwali for their field assistance. We also thank Dr Rafael Argiles Herrero and the Insect Pest Control Section of the International Atomic Energy Agency for their assistance in sourcing materials for the study.

## Author Contributions

**Conceptualization:** Jackson Muyobela, Christian W. W. Pirk, Abdullahi A. Yusuf, Njelembo J. Mbewe, Catherine L. Sole.

**Data curation:** Jackson Muyobela.

**Formal analysis:** Jackson Muyobela.

**Funding acquisition:** Jackson Muyobela, Christian W. W. Pirk, Catherine L. Sole.

**Investigation:** Jackson Muyobela, Christian W. W. Pirk, Abdullahi A. Yusuf, Njelembo J. Mbewe, Catherine L. Sole.

**Methodology:** Jackson Muyobela, Christian W. W. Pirk, Abdullahi A. Yusuf, Njelembo J. Mbewe, Catherine L. Sole.

**Project administration:** Catherine L. Sole.

**Resources:** Jackson Muyobela.

**Supervision:** Christian W. W. Pirk, Abdullahi A. Yusuf, Catherine L. Sole.

**Writing – original draft:** Jackson Muyobela.

**Writing – review & editing:** Jackson Muyobela, Christian W. W. Pirk, Abdullahi A. Yusuf, Njelembo J. Mbewe, Catherine L. Sole.

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
