## [Decision Letter · Decision Letter 0]

31 Mar 2021

Dear Mr Muyobela,

Thank you very much for submitting your manuscript "Standardisation and optimisation of a novel vehicle-mounted sticky trap for the sampling of the savannah tsetse flies Glossina morsitans morsitans Westwood and Glossina morsitans centralis Machado" for consideration at PLOS Neglected Tropical Diseases. As with all papers reviewed by the journal, your manuscript was reviewed by members of the editorial board and by several independent reviewers. The reviewers appreciated the attention to an important topic. Based on the reviews, we are likely to accept this manuscript for publication, providing that you modify the manuscript according to the review recommendations. 

Sincerely,

Philippe Solano

Associate Editor

Jan Van Den Abbeele

Deputy Editor

Reviewer's Responses to Questions

**Key Review Criteria Required for Acceptance?**

**Methods**

-Are the objectives of the study clearly articulated with a clear testable hypothesis stated?

-Is the study design appropriate to address the stated objectives?

-Is the population clearly described and appropriate for the hypothesis being tested?

-Is the sample size sufficient to ensure adequate power to address the hypothesis being tested?

-Were correct statistical analysis used to support conclusions?

-Are there concerns about ethical or regulatory requirements being met?

Reviewer #1: Please see attached document

Reviewer #2: -Are the objectives of the study clearly articulated with a clear testable hypothesis stated?

Yes, the authors state that they aim to evaluate a vehicle-based sampling method against the 'traditional' fly round. The Introduction misrepresents some of the disadvantages of traps. The cost estimate is not for sampling but for control operations. Many traps can be deployed by a team and the trap can operate without humans being present whereas a flyround requires humans and possibly cattle to be present. A survey using a vehicle has all the costs of operating a vehicle. Nonetheless, there is merit in exploring better mobile samping methods for G. morsitans subspecies.

-Is the study design appropriate to address the stated objectives?

I found the description of the experimental design very difficult to follow. It seems that the authors had a number of different transects which they operated simultaneously but it is not clear how they randomised the allocation of sampling methods between transects. In general the language is not very clear throughout the manuscript. I would expect to see some form or Latin square design of transects x days x sampling methods. Are 'blocks' equivalent to transects? I think a clearer description of the experimental design would help with a graphic of the design in the SI perhaps.

-Is the population clearly described and appropriate for the hypothesis being tested?

The study sites are well described and the catches are adequate for comparing the different sampling methods.

-Is the sample size sufficient to ensure adequate power to address the hypothesis being tested?

Yes the catches of tsetse are sufficient to test the hypotheses. However, each of the experiments was conducted over only 2-3 days. For instance the comparison of the fly round and VST was conduced over two days only. It is unclear to me whether treatments have been truly randomised and that 'blocks' are independent.

-Were correct statistical analysis used to support conclusions?

The analyses are poorly described. Did you use a glm or glmm? How did you account for days, transects and treatments? The data for all the experiments are contained in Table 1 but the results of statistical analyses are not presented here. The analyses are presented in verbose and repetitive text. I would much prefer to see these presented in a table with mean catches, CIs and contrasts. The authors refer to a catch index but this is not explained in the methods.

-Are there concerns about ethical or regulatory requirements being met?

The study meets all the expected ethical and regulatory requirements.

**Results**

-Does the analysis presented match the analysis plan?

-Are the results clearly and completely presented?

-Are the figures (Tables, Images) of sufficient quality for clarity?

Reviewer #1: Please see attached document

Reviewer #2: -Does the analysis presented match the analysis plan?

Probably, but the presentation is very difficult to follow. I would prefer to see the mean catches, CIs, catch indices and P values presented in a Table.

-Are the results clearly and completely presented?

No. See above. The results need to be presented in a Table or figure rather than pages of text.

-Are the figures (Tables, Images) of sufficient quality for clarity?

No. There is a single Table of the results which presents total catches. I would like to see the mean catches, CIs, catch indices and P values presented in a Table and the text comments on the main findings.

**Conclusions**

-Are the conclusions supported by the data presented?

-Are the limitations of analysis clearly described?

-Do the authors discuss how these data can be helpful to advance our understanding of the topic under study?

-Is public health relevance addressed?

Reviewer #1: Please see attached document

Reviewer #2: -Are the conclusions supported by the data presented?

The data suggest that tsetse can be caught on vehicle-mounted panels, and that the vehicle mounted devices catch more tsetse than a traditional fly round. The results (6 pages) and discussion are verbose (6 pages) for findings that could be clearly and succinctly presented in a single table. 

-Are the limitations of analysis clearly described?

There is discussion of the limitations of the method but the authors ignore the costs of using a vehicle to monitor tsetse compared to use of traps.

-Do the authors discuss how these data can be helpful to advance our understanding of the topic under study?

The authors suggest that vehicle-counted traps could be used to monitor G. morsitans in tsetse control operations.

-Is public health relevance addressed?

Yes, the authors relate their aims and findings to the control of African trypanosomiasis.

**Editorial and Data Presentation Modifications?**

Reviewer #1: Please see attached document

Reviewer #2: The manuscript is verbose and rambling and the language unclear. I strongly suggest that the authors reduce the overall length, improve the presentation of results and the text highlights the main findings.

**Summary and General Comments**

Reviewer #1: Please see attached document

Reviewer #2: (No Response)

PLOS authors have the option to publish the peer review history of their article (what does this mean?). If published, this will include your full peer review and any attached files.

Reviewer #1: Yes: Glyn Vale

Reviewer #2: No

Figure Files:

Data Requirements:

Reproducibility:

References

---

## [Decision Letter · Decision Letter 1]

1 Jun 2021

Dear Mr Muyobela,

Thank you very much for submitting your manuscript "A novel vehicle-mounted sticky trap; an effective sampling tool for savannah tsetse flies Glossina morsitans morsitans Westwood and Glossina morsitans centralis Machado" for consideration at PLOS Neglected Tropical Diseases. As with all papers reviewed by the journal, your manuscript was reviewed by members of the editorial board and by several independent reviewers. The reviewers appreciated the attention to an important topic. Based on the reviews, we are likely to accept this manuscript for publication, providing that you modify the manuscript according to the review recommendations. 

Sincerely,

Philippe Solano

Associate Editor

Jan Van Den Abbeele

Deputy Editor

Reviewer's Responses to Questions

**Key Review Criteria Required for Acceptance?**

**Methods**

-Are the objectives of the study clearly articulated with a clear testable hypothesis stated?

-Is the study design appropriate to address the stated objectives?

-Is the population clearly described and appropriate for the hypothesis being tested?

-Is the sample size sufficient to ensure adequate power to address the hypothesis being tested?

-Were correct statistical analysis used to support conclusions?

-Are there concerns about ethical or regulatory requirements being met?

Reviewer #1: (No Response)

Reviewer #2: (No Response)

**Results**

-Does the analysis presented match the analysis plan?

-Are the results clearly and completely presented?

-Are the figures (Tables, Images) of sufficient quality for clarity?

Reviewer #1: (No Response)

Reviewer #2: (No Response)

**Conclusions**

-Are the conclusions supported by the data presented?

-Are the limitations of analysis clearly described?

-Do the authors discuss how these data can be helpful to advance our understanding of the topic under study?

-Is public health relevance addressed?

Reviewer #1: (No Response)

Reviewer #2: (No Response)

**Editorial and Data Presentation Modifications?**

Reviewer #1: (No Response)

Reviewer #2: (No Response)

**Summary and General Comments**

Reviewer #1: General

The tabular format in which the authors offer their responses to comments is excellent. 

I agree with all comments made by the other reviewer, ie, Reviewer #2. This is hardly surprising since in essence his/her comments are much the same as mine.

As regards the authors’ response to my comments, I note that the authors have agreed with most of them and made the appropriate changes in almost all instances. Of the comments with which the authors do not agree, I am happy in most cases to regard our divergent stances as matters of opinion which can be tolerated. Of the few remaining differences of opinion, none need preclude publication of the revised MS if the authors wish to continue to disregard the views I offer, but it is my duty to emphasize what those views are, as below.

Divergent opinions

My comments in this section are headed according to the line numbers mentioned in the authors’ table of responses.

145. (Maps) – The clearest, simplest and most concise means of impressing the reader with the extent of the infestations is to indicate in the text the sqkm covered – as indeed the authors have already done. Heaven help us and our readers if every time an important insect is discussed it is necessary to produce also a map of where the creature occurs. Likewise, in most papers we hardly need a map of where behavioural studies were performed – a line of text and a map reference would do. Moreover, the maps of the meandering paths used in each study area do not tell the reader much. It would have been more interesting to have shown the vegetation along the transects, especially since the authors indicate that the vegetation is not homogeneous in some places. Hence, while agreeing to the continued inclusion of the maps, I maintain my opinion that their main benefit is that of adding a bit more colour and packing to an otherwise skimpy MS. 

393 t0 406. (Habitats) – The authors’ proposition seems to be that tsetse will not fly in the direction of unfavourable habitat despite having a bait to explore. That proposition is difficult to square with the fact that tsetse are shown to visit a bait that is travelling in an inhospitable area along the edge of good woodland habitat. It is a pity that the authors made no serious attempt to substantiate this proposition experimentally. The catches from a screen coloured on both sides, with sticky deposit on both the woodland and dambo sides, or on each side separately, might have been pursued more fully to add greater substance and interest to the MS. 

445. (Perfect sticky deposit?) – The authors seem to claim that because the film they used was a sticky product of Rentokil it is reasonable to assume that all flies that touch the screen get caught. This deserves some supporting evidence, but none is offered. Instead, all the reader gets is the anecdotal and unsurprising indication that if a fly gets well stuck it remains stuck. The point at issue is how well the fly gets stuck at the first touch.

453. (Further praise of sticky deposits) – The authors have not answered the question I posed. They state, simply and baldly, that “Most were captured”. Maybe they have no answer to offer.

Specific comments on the revised MS

I had hoped that my comments on the linguistic aspects of the first version of the MS would have given the authors the heads-up about their writing style. Sadly, the authors seem determined to maintain their appallingly casualness. It makes commenting very tedious and STRESSFUL for busy reviewers who are obliged to produce their comments over the weekend. Here are some suggested improvements. Many others could be found.

The line numbers used below refer to the lines of the particular document that shows in full the deletions and additions that the authors have made to the original MS. 

32. – Delete “fly”.

34. – Zambia comma.

39. – differenceS; “or” not “and”?

45. – , respectively comma.

50. – insert “m.” in G. m. morsitans, and italicize the name. Is the reader to believe that the VST is used to sample the BFR?

54. – G. m. morsitans.

56. – “its” not “theirs”.

91. – G. m. morsitans.

95. – Delete “Map showing the “. We can see that it is a map.

106. – Delete second comma.

118. “lower” implies “relative”. If you must use “relative”, what about: “relatively low cost”?

120. – tsetse’s. delete “fly”.

123. – It is strange to say that a device which samples activity can misrepresent activity patterns. It would be good to give a direct reference to the point, rather than the reviews offered.

123. – Split infinitive. 

125. – often makes their use prohibitive. You cannot imply that the use is always prohibitive when, in reality, the traps are often used for large-scale surveys. 

140. – morsitans comma, particularly AT low population (No S) densities.

141. – Delete “set out to”.

142. – delete first “and”.

143-144. – to identify the optimal colour and orientation of the trap and to assess ….

145. – Why keep giving the name in full, together with its acronym, when you have already explained the acronym? 

146. – The same type of booboo as above is made again in respect of the BFR.

159. – Glossina m. morsitans.

164-165 and 174-175 -- in each case a reference to the diet would not be amiss.

166 and 176. – km.

I78. – Two fabrics were used in trap construction. These were …..

180. – They were cut ….

190-196. -- I doubt the story offered here. The spectral reflectance of the cloth might not be affected much by the film, but the point at issue is the reflectance of the film. Maybe, as the authors say, the film absorbs UV light, and maybe the cloth absorbs all or most of the non-UV light that gets through the film, but that says little about the non-UV light that is reflected by the film. Fig. 3 shows clearly that white light visible to humans is reflected by the film, and such light contains wavelengths known to be perceptible to tsetse. The upshot is that the light coming from the black and/or blue panels coated with film is not the same as the light from panels that are not so coated.

197. --Butanone and 1-octen-3-ol were used as attractants, according to ….

208. – “completely randomized block design experiments” is about as ugly as it gets.

209. -- In the same vein, “best panel trap colour design” is vague and stressful. For example, does it mean the colour of the best panel trap, or the best colour of the panel trap. The whole sentence needs reorganizing and I hope I can leave that to persons who will know better than I what the sentence is trying to say.

212. – outlined. 

213. – km.

215. -- Does this mean that there were two trucks, each operating simultaneously?

2i7. – Means what? 

218. – Does this mean that there was one panel, smaller than the overall size of the trap, and that the panel was moved around at random to occupy different parts of the trap surface?

212. – Inserting “An” before “octenol” would make it immediately clear that “octenol” is used as an adjective, not a noun.

225. – km.

227-228. – comma respectively. Does a traverse have a temperature and humidity?

229. – DEARY ME! It is said in the table of responses that all instances of “Km” have been changed to “km”, but in fact nothing of the sort has been done.

NB. – I am going to stop reviewing the dreadfully cavalier English of the Methods and Results from this point onwards. I just do not have the time and patience to wade through it all. One or more of the most careful of the authors must take a serious and determined look at it all. I will begin reading again from the start of the Discussion.

376. – “vehicle-mounted all-blue, all-black, and blue-black sticky panel traps” is hardly clear.

378. – Catches of sticky traps depend on a sequence of three things: (i) attraction of flies from a distance, (ii) close range orientation towards the sticky surface, (iii) alighting on the surface in a manner that is likely to ensure permanent retention. Is the intention to say that all of these separate responses are not affected by colour. If so, I fear that the intention is awry, and in any case the idea is contradicted in the next sentence. It makes for very confused reading. 

388. – The sentence starting here is of vague meaning in several ways. For example, if one considers the ability of a device to elicit a landing response it would make most sense to assess what proportion of flies near the bait are encouraged to alight. As matters stand, all that is measured is how many flies alight and get stuck with different treatments. It could well be that certain treatments encourage attraction to the vicinity of the bait and also reduce the strength of the alighting response. Strictly speaking, the results offer no reason to suggest that translocation of panels encourages alighting behaviour. 

390. -- Italics for G. m. morsitans.

396. – objectives ARE.

418-419 . – known to be; on the panel. 

433. – Split infinitive.

436. -- Differences BETWEEN.

443. – Delete “among other host seeking stimuli” (stimuli do not seek hosts). 

444-445. – Perhaps the differences between the geometry of the habitats of the two species is …..

448. -- Italics for G. morsitans.

451. Given the skimpy nature of the present MS, I would have thought it safer to say that more work is recommended to confirm the claim that there is indeed a “drastic” difference in the olfactory responses of the two sub-species.

466. No evidence has been adduced to support the claim that ”all landing flies are secured”.

477 and 478. – km.

487. – Delete “flies”.

503. – more rapid and efficient than the BFR for sampling G. morsitans, so making it … 

References. – Numbers 34 and 50 would look better without the excessive capitalization

Reviewer #2: The authors have addressed my previous comments. I think the manuscript would benefit from editing for language since it is still verbose and unclear in places. I notice that the first reviewer has made detailed comments on the language and I urge the authors to follow the reviewer's advice. 

I recommend that the authors re-consider their use of the word 'efficient' and 'efficiency'. In most cases they are reporting that catches increased and this is not necessarily due to imporvements in the efficiency of the device. I notice that the first reviewer makes a similar comment in their first review but the authors seem to have ignored this comment.

I am not convinced that the results allow the authors to infer much about the host-oriented behaviour of tsetse. The device is mounted on a large vehicle which itself presents a large visual target but the discussion of tsetse behaviour seems to assume that the responses are to the panels only. I think the main value of the paper is that it highlights the utility of vehicle-mounted sticky screens for sampling tsetse.

I am not convinced by the arguments that this method is more cost-effective than using traps. The authors ignore several papers by Shaw et al that show that entomological modelling using traps is much cheaper than the $283/km2 they report.

I would like to see the various statements that flyrounds are 'the' recommended method for sampling G. morsitans changed. I suggest they write that flyrounds are 'a' method used for sampling G. morsitans. Traps are used widely and I suspect that a systematic review of the literature would show that they are more widely used than flyrounds.

I recommend that the article is accepted for publication subject to addressing the above points.

PLOS authors have the option to publish the peer review history of their article (what does this mean?). If published, this will include your full peer review and any attached files.

Reviewer #1: No

Reviewer #2: No

Figure Files:

Data Requirements:

Reproducibility:

References

---

## [Editor Report · Decision Letter 2]

2 Jul 2021

Dear Mr Muyobela,

We are pleased to inform you that your manuscript 'A novel vehicle-mounted sticky trap; an effective sampling tool for savannah tsetse flies Glossina morsitans morsitans Westwood and Glossina morsitans centralis Machado' has been provisionally accepted for publication in PLOS Neglected Tropical Diseases.

Best regards,

Philippe Solano

Associate Editor

Jan Van Den Abbeele

Deputy Editor

---

## [Editor Report · Acceptance letter]

14 Jul 2021

Dear Mr Muyobela,

We are delighted to inform you that your manuscript, "A novel vehicle-mounted sticky trap; an effective sampling tool for savannah tsetse flies Glossina morsitans morsitans Westwood and Glossina morsitans centralis Machado," has been formally accepted for publication in PLOS Neglected Tropical Diseases.

Best regards,

Shaden Kamhawi

co-Editor-in-Chief

Paul Brindley

co-Editor-in-Chief
